# LATENT PLANNING VIA EMBEDDING ARITHMETIC: A CONTRASTIVE APPROACH TO STRATEGIC REASONING

## ABSTRACT

Planning in high-dimensional decision spaces is increasingly being studied through the lens of learned representations. Rather than training policies or value heads, we investigate whether planning can be carried out directly in an evaluation-aligned embedding space. We introduce SOLIS, which learns such a space using supervised contrastive learning. In this representation, outcome similarity is captured by proximity, and a single global *advantage vector* orients the space from losing to winning regions. Candidate actions are then ranked according to their alignment with this direction, reducing planning to vector operations in latent space. We demonstrate this approach in chess, where SOLIS uses only a shallow search guided by the learned embedding to reach competitive strength under constrained conditions. More broadly, our results suggest that evaluation-aligned latent planning offers a lightweight alternative to traditional dynamics models or policy learning. All source code and pretrained models will be made available upon publication.

## 1 INTRODUCTION

Planning in high-dimensional state spaces remains a central challenge in artificial intelligence. Traditional reinforcement learning methods rely on explicit search or learned policies, but these can become inefficient when the space of possible actions grows large (Jiang et al., 2023; Gieselmann and Pokorny, 2022). A recent line of work has explored latent planning, which reasons over compact latent representations rather than raw states, thereby reducing complexity and enabling more human-like decision-making (Hafner et al., 2019; 2020; Schrittwieser et al., 2020).

We explore whether a learned embedding space can directly support planning by treating decision-making as interpolation between regions of advantage. To this end, we introduce **SOLIS** (*Strategic Optimization via Latent Interpolative Spaces*), a chess engine built on a transformer encoder trained with supervised contrastive learning to embed states into an evaluation-aligned latent space. In this space, distance correlates with similarity of outcome, and directional movement corresponds heuristically to progress toward favorable states. SOLIS utilizes this by ranking candidate moves according to their alignment with a single global "advantage vector" that orients the latent space from losing to winning regions.

Even at modest search depths, SOLIS achieves strong results. With only a 5-ply search under 50 ms/move blitz conditions, it achieves an Elo of 2500+; in equal-depth matches at search depth $S \in \{3, 4, 5\}$ with a fixed branching factor, it is competitive with a configured Stockfish baseline in our setup. Visualizations of real games further show that trajectories trace smooth paths through the embedding space, offering qualitative interpretability.

In contrast to latent planning methods that couple learned dynamics with policy or value heads (Hafner et al., 2019; 2020; Schrittwieser et al., 2020), SOLIS relies solely on the representation of an evaluation-aligned space for action selection. Although we use chess as a testbed, the vector-based latent planning approach may extend to other perfect-information domains where state evaluations are available or learnable.

## 2 BACKGROUND

Planning in very large action spaces presents a formidable challenge in artificial intelligence because the number of potential action sequences grows exponentially with every additional decision step. This requires careful modeling and thoughtful critical thinking about building a robust methodology. This section discusses important details about how our proposed model operates.

### 2.1 MARKOV DECISION PROCESSES

Sequential decision making tasks are formalized as Markov Decision Processes (MDPs), which take the form of a tuple $\{\mathcal{S}, \mathcal{A}, T, R, p(s_0), \gamma\}$, where $\mathcal{S}$ denotes the state space, $\mathcal{A}$ the action space, $T$ the transition model, $R$ the reward function, $p(s_0)$ the initial state distribution, and $\gamma \in [0,1]$ the discount factor for future rewards (Puterman, 2014). A policy is defined as $\pi : \mathcal{S} \to p(\mathcal{A})$, a mapping from states to probability distributions over actions.

In this setting, a *model* refers to reversible access to the MDP dynamics: given a state–action pair, the model predicts the next state and reward (Moerland et al., 2022). For games such as chess and Go, the rules of the game provide a direct model. In domains where the rules are not known, the model must be approximated from data (Gieselmann and Pokorny, 2022).

Planning and reinforcement learning can then be distinguished by how they use this access to MDP dynamics (Moerland et al., 2022). Planning methods assume access to a model and simulate from the current state to choose an action, similar to how humans plan in their minds (Russell and Norvig, 1995). Because the solution is relative to the current state, it is considered *local* and is thus discarded after execution. Reinforcement learning methods, in contrast, typically lack reversible access to the MDP dynamics (Sutton and Barto, 1998). They must interact with the environment step by step, and as a result, they learn a *global solution*, such as a policy or value function, that generalizes across the entire state space.

Model-based reinforcement learning combines both elements by using a model while also learning a global solution. According to Moerland et al. (2022), this can take two forms. One is model-based RL with a learned model (e.g., Dyna (Sutton, 1991)), where both the dynamics and the global solution are learned. The other is model-based RL with a known model (e.g., AlphaZero (Silver et al., 2017)), where the dynamics are given and only the global solution is learned.

#### 2.1.1 MODEL-BASED RL WITH A LEARNED MODEL

A large body of work falls into the category of model-based RL with a learned model, where both the dynamics and the global solution are learned. PlaNet (Hafner et al., 2019) learns a latent dynamics model and combines it with a learned global solution. Dreamer (Hafner et al., 2020) and its successors DreamerV2 (Hafner et al., 2022) and DreamerV3 (Hafner et al., 2024) extend this approach by scaling to more complex tasks, while still maintaining global value functions and policies. MuZero (Schrittwieser et al., 2020) also learns a latent dynamics model, but integrates it with tree search to strengthen planning while training a global policy and value function. Other examples include Dyna (Sutton, 1991) and MBPO (Janner et al., 2021), both of which use learned models to support value or policy learning.

#### 2.1.2 MODEL-BASED RL WITH A KNOWN MODEL

When the environment dynamics are available, the algorithm can focus on coupling search with learning. AlphaZero (Silver et al., 2017) uses exact game rules as the model, combining Monte Carlo tree search (MCTS) with learned policy and value heads. SAVE (Hamrick et al., 2020) similarly assumes a simulator and integrates Q-learning with MCTS: a learned Q-function serves as a prior for search, while search-produced Q-estimates are amortized back into the network.

#### 2.1.3 LATENT PLANNING

Latent planning simplifies decision-making by operating in a compressed representation space rather than the raw state space. PlaNet (Hafner et al., 2019), Dreamer (Hafner et al., 2020), and MuZero (Schrittwieser et al., 2020) pair latent dynamics models with value or policy heads, enabling

planning through simulations in the learned space. Expansive Latent Space Trees (ELAST) (Gieselmann and Pokorny, 2022) cast latent planning as explorative tree search by combining contrastive representations with sampling-based expansion to address long-horizon control. Trajectory Autoencoding Planner (TAP) (Jiang et al., 2023) takes a similar approach by learning a compact latent action space and planning over discrete action codes.

SOLIS does not train policies or value heads and does not learn a dynamics model; instead it distills evaluation targets into the representation. Planning then proceeds by moving between regions aligned with favorable outcomes, using only the representation space at inference.

## 2.2 CHESS ENGINES

IBM successfully scaled chess computing in 1997 with their Deep Blue engine to defeat the reigning world chess champion, Garry Kasparov, in a head-to-head match. Early versions of Stockfish (Romstad et al., 2008) followed a similar approach to Deep Blue: handcrafted evaluation functions paired with deep alpha-beta tree search. The methodology was compelling and achieved superhuman performance, though it became clear that the limiting factor was the combined chess mastery of those designing the static evaluation function.

The rating bottlenecks associated with static evaluation inspired a vein of research independent of human guidance. AlphaZero (Silver et al., 2017) and its open-source counterpart Leela Chess Zero (Pascutto and Linscott, 2018) made a significant leap to engines that learned to evaluate positions entirely through self-play reinforcement learning, with no prior knowledge beyond the rules of chess. Similarly, modern versions of Stockfish adopted an efficiently updatable neural network evaluation module (NNUE) (Nasu, 2018) to accelerate position evaluation while maintaining traditional alpha-beta search.

While these advances have produced engines that far surpass human players in strength, the underlying search algorithms remain largely unchanged. They still follow principles first proposed by v. Neumann (1928) and Shannon (1950) and later integrated into Deep Blue, relying on deep alpha-beta search through millions of states (Lai, 2015). In this work, we explore whether contrastive learning can enable a more efficient search process by treating planning as traversal through a high-dimensional embedding space.

## 2.3 CONTRASTIVE LEARNING

Contrastive learning of representations encourages a model to embed similar inputs closer together and dissimilar inputs farther apart in a latent space (Chen et al., 2020). More concretely, given a batch $I$ of training examples containing an anchor input $x_i$, an encoder $f(\cdot)$ maps the anchor to a normalized embedding $z_i = f(x_i)$. Positive pairs $(z_i, z_j)$ correspond to semantically similar inputs, while all other pairs of samples in the batch are considered negatives.

A common contrastive objective is InfoNCE (van den Oord et al., 2019), defined as:

$$\ell_{i,j} = -\log \frac{\exp(\text{sim}(z_i, z_j)/\tau)}{\sum_{k \in A(i)} \exp(\text{sim}(z_i, z_k)/\tau)}, \tag{1}$$

where $A(i) = I \setminus \{i\}$ is the set of all samples in the batch excluding the anchor $i$, $\text{sim}(\cdot, \cdot)$ denotes a similarity metric, and $\tau$ is a scalar temperature parameter. In the case of self-supervised contrastive learning, the positive sample $z_j$ is an augmentation of the anchor $z_i$, and labels are not required (Chen et al., 2020; Khosla et al., 2021).

When labels are available, the supervised contrastive loss (SupCon) (Khosla et al., 2021) extends InfoNCE by allowing multiple positive examples per anchor. The SupCon loss is defined as:

$$\mathcal{L}_{\text{sup}} = \sum_{i \in I} \frac{-1}{|P(i)|} \sum_{p \in P(i)} \log \frac{\exp(\text{sim}(z_i, z_p)/\tau)}{\sum_{a \in A(i)} \exp(\text{sim}(z_i, z_a)/\tau)}, \tag{2}$$

where $P(i)$ denotes the set of positive examples corresponding to anchor $i$, and $A(i)$ is defined as above. Increasing the number of positive and negative pairs has been shown to improve representation quality (Khosla et al., 2021; Chowdhury et al., 2024).

# 3 METHODS

We now describe the components of our approach, including tokenization, model architecture, training procedure, and action selection during inference.

## 3.1 INPUT REPRESENTATION AND TOKENIZATION

Each chess position is represented using its Forsyth-Edwards Notation (FEN) ASCII string, which compactly encodes the piece placement, side to move, castling rights, en passant target, half-move clock, and full-move counter. Following the tokenization scheme proposed by Ruoss et al. (2024), we tokenize each FEN into a fixed-length sequence of 77 tokens by expanding run-length encodings. As pointed out by Ruoss et al. (2024), this tokenization makes chess a non-Markovian problem, since information about past states is omitted but could be useful for future actions in special cases (e.g., threefold repetition or the fifty-move rule). This is discussed in more detail in Section 5.

## 3.2 ENCODER ARCHITECTURE

Our encoder is a multi-layer transformer (Vaswani et al., 2023) adapted for tokenized chess positions. We train two variants, SOLIS Base and SOLIS Mini. We summarize the model configurations used in our experiments in Table 1.

Table 1: Transformer model configurations used in our experiments.

| Model | Layers | Embedding dim. $D$ | Heads | MLP Size | Params |
|---|---|---|---|---|---|
| SOLIS Mini | 6 | 128 | 8 | 256 | 0.8M |
| SOLIS Base | 6 | 1024 | 16 | 1024 | 41M |

Each input sequence is embedded into $D$-dimensional vectors through a learned token embedding matrix. A special classification token (CLS) (Devlin et al., 2019; Dosovitskiy et al., 2021) is prepended to the embedded sequence to aggregate information across the input. Because the input sequences are fixed length, we add a learned positional encoding to each token embedding (Ruoss et al., 2024). The resulting sequence is processed by stacked transformer encoder layers with GELU (Hendrycks and Gimpel, 2023) activations and dropout (Srivastava et al., 2014). The final hidden state corresponding to the CLS token is extracted, passed through a linear projection, and $\ell_2$ normalized.

## 3.3 SUPERVISED CONTRASTIVE TRAINING

We train our encoder using 5 million randomly sampled positions from the ChessBench dataset (Ruoss et al., 2024), each annotated with a Stockfish-evaluated win probability for the player to move. We normalize all evaluations to represent White's perspective, such that values near 1.0 indicate a decisive advantage for White and values near 0.0 indicate an advantage for Black.

To define positive samples for training, we set $\delta$ as the evaluation margin for identifying similar positions. For each anchor, we precompute all positions whose win probabilities differ by less than $\delta$ and randomly sample five as positives during training. To build the batch-level mask, we compare the win probabilities of all samples and mark pairs as positive if their evaluation difference is below $\delta$. This margin-based sampling pulls together states of nearly equal evaluation and pushes apart positions with larger gaps. By defining positives as such, the embedding space ultimately encourages an ordering from losing to winning positions.

We apply supervised contrastive learning (SupCon) (Khosla et al., 2021) with the constructed batch masks to define positive and negative pairs. Pairwise scores are computed using cosine similarity, with temperature $\tau = 0.07$ (Girdhar et al., 2023; Radford et al., 2021) controlling the sharpness of the distribution. Optimization uses stochastic gradient descent with momentum 0.9 (Dosovitskiy et al., 2021; Chen et al., 2020; He et al., 2015), batch size 128, for 400,000 steps. Training is carried out on six NVIDIA L40S GPUs (48 GB each) with data parallelism (Paszke et al., 2019). Complete training hyperparameters are presented in Appendix A.2.

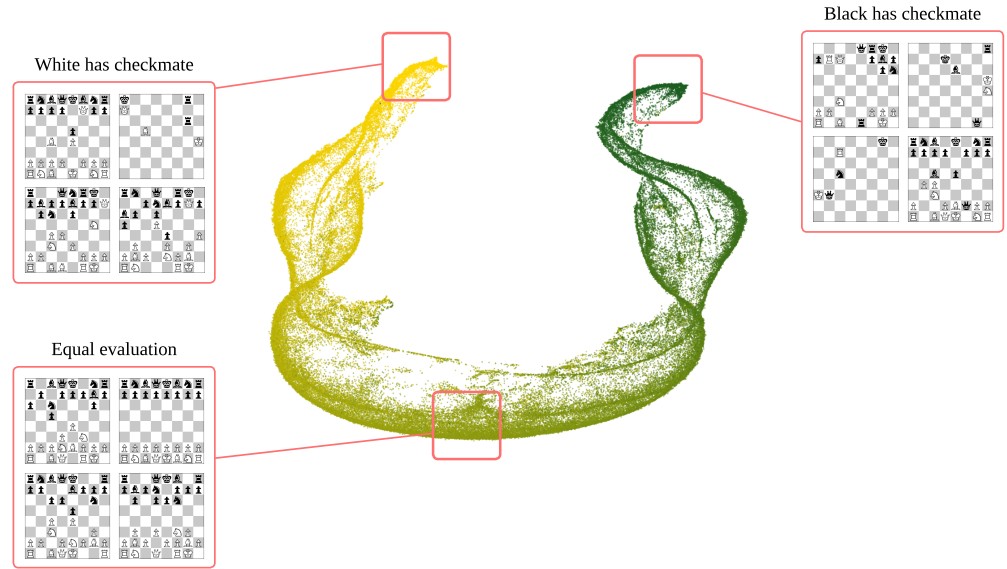

Figure 1: UMAP projection of the learned embedding space of our Base model, colored by win probability (gold = White favored, green = Black favored).

We visualize the learned representation space of our Base model in Figure 1 using UMAP (McInnes et al., 2020) for 2D projection. Each point represents an encoded chess position from a test dataset, colored by its Stockfish-evaluated win probability for White. Positions where White has a strong advantage are colored in gold (*White has checkmate*); positions where Black has a strong advantage are colored in dark green (*Black has checkmate*).

## 3.4 EMBEDDING-GUIDED SEARCH

At inference we plan directly in the learned space by ranking legal continuations using a single global advantage direction. We compute means from a held-out dataset at the extremes of evaluation,

$$\mu_{\text{White}} = \mathbb{E}[\, z \mid p{=}1.0\,], \qquad \mu_{\text{Black}} = \mathbb{E}[\, z \mid p{=}0.0\,],$$

and define the normalized advantage vector

$$\vec{a} = \frac{\mu_{\text{White}} - \mu_{\text{Black}}}{\|\mu_{\text{White}} - \mu_{\text{Black}}\|}.$$

In our experiments, we consider two mechanisms to convert this vector into scalar scores.

### 3.4.1 UNANCHORED (GLOBAL-DIRECTIONAL)

The first approach scores each child embedding directly by its alignment with the advantage direction. Each child embedding $z'$ is scored by

$$\text{score}_{\text{dir}}(z') = \langle z', \vec{a} \rangle,$$

favoring moves whose embeddings extend furthest along the advantage direction regardless of the origin. This is the scoring mechanism shown in Figure 2 for illustration.

### 3.4.2 ANCHORED (RELATIVE PROJECTION)

Alternatively, we can shift by the Black mean and score

$$\text{score}_{\text{anch}}(z') = \langle z' - \mu_{\text{Black}}, \vec{a} \rangle,$$

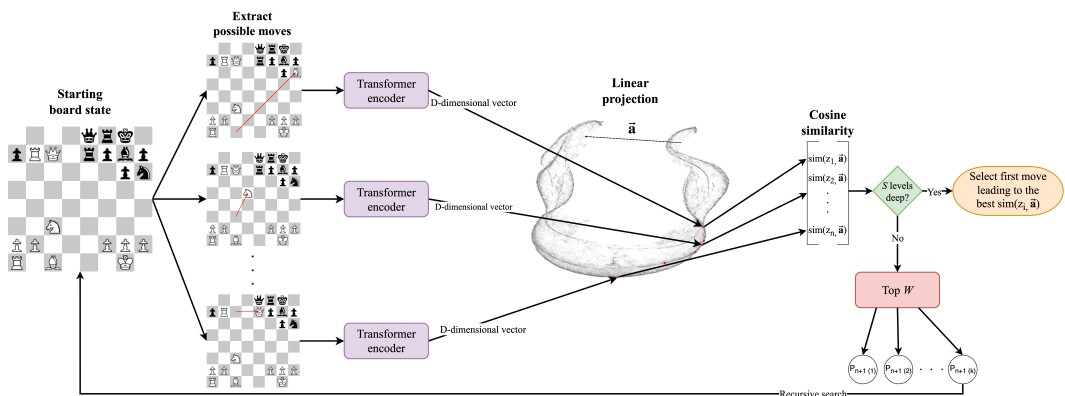

Figure 2: System overview of SOLIS. Candidate moves are embedded and scored by their alignment with the global advantage vector. For simplicity, this diagram shows the unanchored scoring mechanism $\cos(z', \vec{a})$.

so that the value is proportional to signed distance from $\mu_{\text{Black}}$ along $\vec{a}$. This removes dependence on an arbitrary origin, and in practice provides a more stable basis for comparing continuations.

The search process (Figure 2) is the same for both scoring mechanisms. From the *"Starting board state"* we *"Extract possible moves"* and embed each child in parallel with the *"Transformer encoder"*. We then compute either $\text{score}_{\text{dir}}$ or $\text{score}_{\text{anch}}$ in the *"Cosine similarity"* block and select the *"Top-W"* children. If we have reached *"S levels deep?"* the procedure terminates and we *"Select first move..."*; otherwise we continue the *"Recursive search"* with min–max preference (White maximizes, Black minimizes). We employ Zobrist hashing (Zobrist, 1990) for a transposition table to avoid re-evaluating positions.

## 4 RESULTS

### 4.1 RATING CALCULATION

To estimate Elo ratings, each model was evaluated against Stockfish with a fixed per-move time limit of 50 ms (Ruoss et al., 2024). Full Stockfish configurations are provided in Appendix A.3. We performed an ablation study over both the breadth and depth of the search. At each depth/width configuration, our models played 600+ games with alternating colors against Stockfish configured at different Elo caps. Both planning processes (unanchored and anchored) were evaluated under this setup. Caps were selected to ensure that each model configuration yielded at least one matchup with a positive win rate and one with a negative win rate. Elo ratings and 95% CIs were estimated using Bayesian logistic regression via the BayesElo (Coulom, 2008) program with the default confidence parameter of 0.5. Elo estimates for the unanchored planning method are presented in Figure 3; estimates for the anchored method are shown in Figure 4. Stockfish rating caps and detailed match results between SOLIS and Stockfish are presented in Appendix A.6 for Elo reproducibility.

**Unanchored ratings** With unanchored planning, Elo ratings scale consistently with search. Depth provides the most considerable gains, while width shows a clear jump from $W \in \{1, 2\}$ to $W=3$ but little improvement beyond that at $W=5$. This plateau indicates that the strongest continuations are generally within the top three candidates, and expanding further adds little strength. At shallow search depths, SOLIS Base outperforms Mini by roughly 100–250 Elo; at depths 4 and 5, Mini is competitive with Base and even outperforms it with $W \in \{3, 5\}$ for this planning mechanism.

**Anchored ratings** With anchored scoring, Elo estimates rise smoothly with depth and width, and the gap between Base and Mini remains in the 100–250 range. Both models exceed 2500 Elo at depth 5, but unlike in the unanchored case, the Base model is consistently stronger than Mini across all settings. Anchoring also delivers a steady 10–30 Elo gain for Base, suggesting that shifting relative to the Black mean provides a more reliable basis for evaluation.

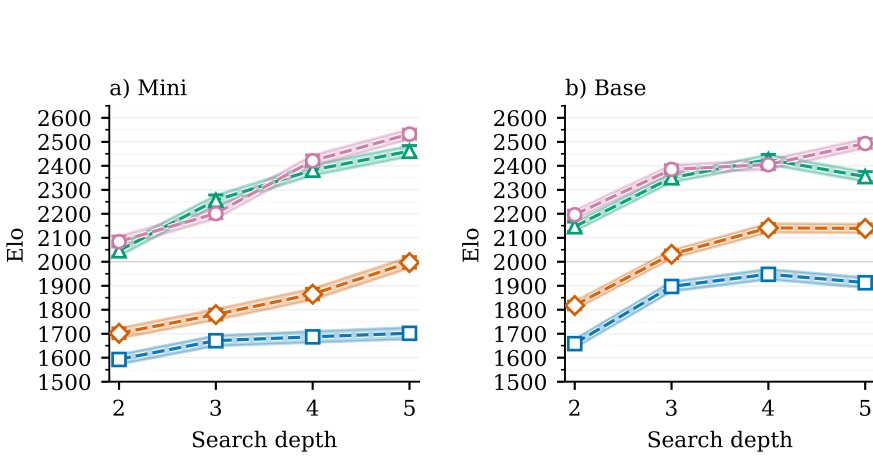

Figure 3: Elo calculations for our Mini and Base models at varying search widths and depths with the *unanchored* planning method. Shaded bands denote 95% confidence intervals.

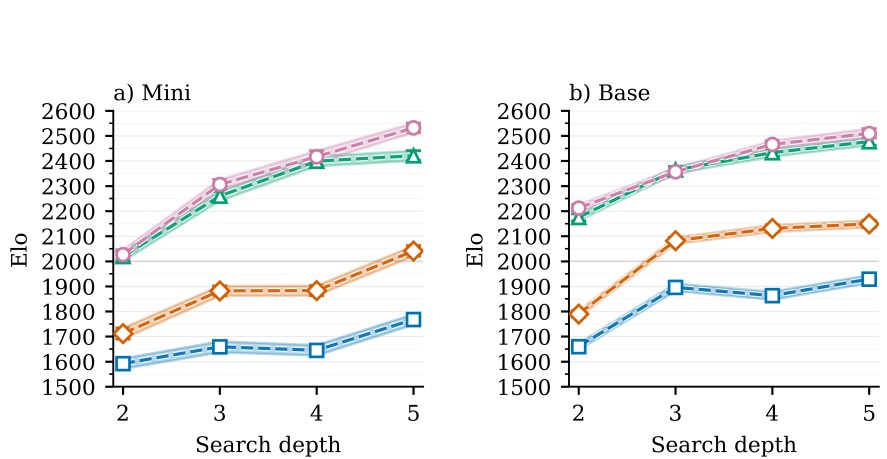

Figure 4: Elo calculations for our Mini and Base models at varying search widths and depths with the *anchored* planning method. Shaded bands denote 95% confidence intervals.

## 4.2 EQUAL-DEPTH SIMULATIONS

We evaluate SOLIS in head-to-head matches against Stockfish at fixed search depths. Based on the Elo scaling experiments in Section 4.1, we fix the branching factor at $W=3$. Wider searches revealed only marginal gains beyond this point, indicating that the strongest continuations are generally among the top three candidates. Stockfish, in contrast, uses its default expansion with no fixed branching factor. At each depth $S \in \{3, 4, 5\}$ we simulate 200 games with alternating colors. Results for unanchored and anchored scoring are shown in Table 2.

**Simulation results**   With unanchored scoring, SOLIS Base achieves higher match points than Stockfish across all tested depths, while Mini performs comparably at depths 3 and 5 but trails at depth 4. With anchored scoring, Base loses narrowly at depth 3 but outperforms Stockfish at depths 4 and 5, and Mini records wins at depths 3 and 5 but again falls behind at depth 4.

Table 2: Match points from SOLIS vs. Stockfish. The middle portion shows results when planning is performed by direct similarity with the global advantage vector. The rightmost portion shows results when planning is performed by anchored projection. Higher is better; draws count as 0.5.

| Config | Depth | Direct Similarity | | Anchored Projection | |
|---|---|---|---|---|---|
| | | SOLIS score | Stockfish score | SOLIS score | Stockfish score |
| Base | 3 | **106** | 94 | 98 | **102** |
| | 4 | **109** | 91 | **108** | 92 |
| | 5 | **107.5** | 92.5 | **110** | 90 |
| Mini | 3 | **108** | 92 | **105** | 95 |
| | 4 | 85 | **115** | 91 | **109** |
| | 5 | **101** | 99 | **113** | 87 |

**Nodes searched**  For SOLIS with a fixed branching factor of $W=3$, the maximum nodes searched at depth $S$ is $N(S) = \sum_{i=0}^{S} 3^i$, giving $N(3) = 40$, $N(4) = 121$, and $N(5) = 364$. By comparison, Stockfish explores a variable number of nodes per move depending on pruning and heuristics. At depth 3 it searched an average of 137 nodes (maximum 1625), at depth 4 it averaged 214 nodes (maximum 4463), and at depth 5 it averaged 338 nodes (maximum 10611). Despite searching far fewer nodes, SOLIS is competitive with this Stockfish configuration, indicating that **much of the strength normally gained from broad expansion can be recovered with higher-quality representations.**

### 4.3 INTERPRETABLE GAME TRAJECTORIES

We visualize latent sequences of moves from real games to emphasize the interpretability of the embedding space. Figure 5 shows three representative examples: a game won by White, a draw that remains balanced throughout, and a game won by Black. Each sequence is plotted over the same 2D projection from a sample dataset. Positions are embedded independently and connected with arrows to indicate the progression of movement. Games where one player decisively wins tend to follow smooth paths through the space, while closely contested games fluctuate around the center. These trajectories complement the earlier embedding visualization (Figure 1) by depicting how the planning process translates into movement toward the regions corresponding to winning or losing outcomes.

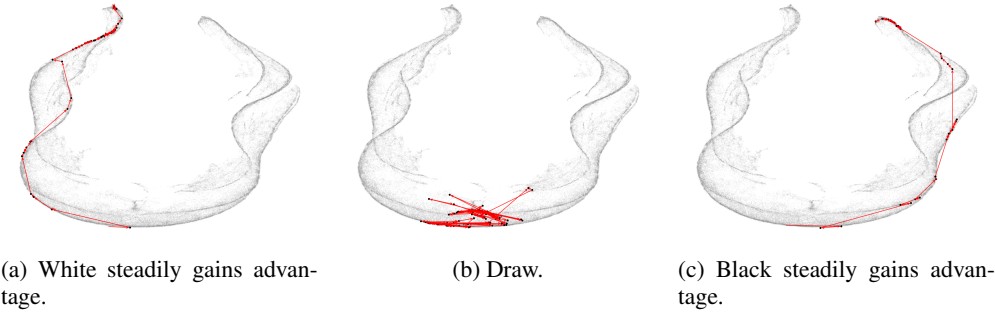

(a) White steadily gains advantage.

(b) Draw.

(c) Black steadily gains advantage.

Figure 5: Latent trajectory visualizations of three games embedded in the shared representation space. Red arrows indicate the progression of positions as the game unfolds.

## 5 DISCUSSION

### 5.1 LIMITATIONS

**Blindness to threefold repetition**  As discussed, SOLIS makes moves based on alignment with an advantage vector, resulting in effective decision-making with minimal overhead. Part of this

simplistic approach, though, means disregarding move history, which can lead to irrational decision-making. Most notable is blindness to threefold repetition: if the same position (including side to move, castling and en-passant rights) occurs three times at any points in the game, either player can claim a draw. A savvy opponent can engineer repeats (e.g., perpetual checks) to neutralize winning positions that SOLIS would otherwise convert. Game *C* in Appendix A.4 shows an example where SOLIS reaches a winning position against Stockfish but allows a premature draw by repetition.

**Indecisiveness in winning positions**   SOLIS cannot distinguish between forcing positions of different lengths. For example, both mate-in-1 and mate-in-20 have a win probability of 1.0, though the shorter line should be preferred. When SOLIS selects the longer continuation, it leaves room for the opponent to complicate the position, which can occasionally result in unnecessary draws or even losses. Ruoss et al. (2024) encountered the same issue and sidestepped it by consulting Stockfish as an external oracle. SOLIS is opting to progress towards its own solution, but this is still a gap to be filled. Game *I* in Appendix A.5 shows a case where SOLIS has many winning continuations for several moves in a row, but makes suboptimal moves leading to a draw.

## 5.2   Conclusion

We studied whether planning can be done directly in an evaluation-aligned latent space instead of training a policy or value head. We introduced SOLIS, which learns a contrastive embedding where proximity reflects outcome similarity and a single global *advantage vector* points from losing to winning regions. Planning then reduces to vector arithmetic by ranking candidate actions by alignment with this direction and expanding a small search.

In chess, SOLIS uses a fixed branching factor $W=3$ and shallow depths $S \in \{3, 4, 5\}$. Across these settings, it matches or exceeds a time-capped Stockfish in several equal-depth matchups while searching far fewer nodes per move (40–364 for SOLIS versus hundreds to thousands for Stockfish at the same depths). Elo rises with depth; with anchored scoring, both Mini and Base exceed 2500 Elo at depth five, and anchoring gives a consistent 10–30 Elo gain for Base. Width brings little benefit past three candidates, which suggests that most strength comes from the quality of the learned space rather than broad expansion. The embedding also yields readable game paths: decisive games move steadily toward winning regions, while draws remain near neutral.

These results support evaluation-aligned latent planning as a simple and compute-efficient alternative to rollout-based search or explicit dynamics models. We expect the same idea to extend to other perfect-information games and to control tasks with large action spaces, where low-branching search is valuable. Future work includes adding rule-aware state features, preferring shorter forced wins, combining SOLIS with policy priors or MCTS, studying calibration and uncertainty, and testing transfer beyond chess. Code and pretrained models will be released upon publication.

## Reproducibility Statement

The full training setup, including hyperparameters and implementation details, is provided in Appendix A.2. Section 3 describes the encoder architecture, input representation, supervised contrastive objective, and inference procedure, with model configurations summarized in Table 1. Evaluation details, including match setup, Stockfish configurations, and rating calculations, are given in Section 4 and Appendix A.3. Detailed match results are included in Appendix A.6 for Elo verification. All source code and pretrained models will be released upon publication.

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

## A APPENDIX

### A.1 USE OF LARGE LANGUAGE MODELS (LLMS)

Large language models were used for minor tasks, including formatting and figure/table placement. They were not used to design experiments, analyze results, or generate technical content. All methodological and experimental contributions were developed and verified only by the authors.

## A.2 TRAINING SETUP

We train both SOLIS configurations (Mini, Base) with supervised contrastive learning on 5M Chess-Bench positions, using cosine similarity with $\ell_2$-normalized embeddings and SGD with momentum on 6 48 GB L40S GPUs for 400k steps (batch size 128).

Among these settings, $\tau$ and $\delta$ are the most consequential. The temperature $\tau = 0.07$ follows prior contrastive work (Radford et al., 2021; Girdhar et al., 2023), where smaller values sharpen similarity scores and larger ones oversmooth gradients. We trained with $\tau \in \{0.05, 0.07, 0.1\}$ and found 0.07 converged most reliably across model sizes. The margin $\delta = 0.05$ balances collapsing too many positions into one neighborhood (large $\delta$) against fragmenting the space (small $\delta$), and also reflects Stockfish evaluation noise, since sub-0.05 differences are rarely meaningful in practice.

Table 3: SOLIS training hyperparameters.

| Attribute | Value |
|---|---|
| Loss | SupCon |
| $\tau$ | 0.07 |
| $\delta$ (positive margin) | 0.05 |
| Positives per anchor | 5 |
| Optimizer | SGD |
| Momentum | 0.9 |
| Learning rate | 0.05 |
| Batch size | 128 |
| Steps | 400k |
| Dropout | 0.10 |
| Activation | GELU |
| Similarity | cosine |
| Embedding norm | $\ell_2$ |
| Token length | 77 |

## A.3 STOCKFISH SETUP

For all experiments, we use the official Linux release of Stockfish 16 (Romstad et al., 2008) with a hash size of 32 MB and default threading. Rating caps are configured using its `UCI_LimitStrength` and `UCI_Elo` settings; depths are configured similarly using the `depth` parameter. To calculate nodes searched for a given move, we set the engine information level to `INFO_ALL` and extract the `nodes` attribute.

We provide sample games between SOLIS and Stockfish, with SOLIS limited to search depth $S$ and width $W$, and Stockfish with unbounded search but a rating cap. Games *A*, *B*, and *E* display SOLIS' tactical understanding and long-term planning. Games *C* and *D* are representative examples of SOLIS' potential weaknesses. The left portion of each table encodes the moves of the game in Portable Game Notation (PGN) format; the right portion of each table displays a critical moment in the game.

```
1.  d4 Nf6 2.  c4 d6 3.  Nc3 e5 4.  Nf3 Qe7 5.  e4 c6
6.  d5 Qc7 7.  Be2 h6 8.  a4 a5 9.  O-O Be7 10.  h3
Na6 11.  Be3 Nb4 12.  Kh1 Nh7 13.  Qd2 O-O 14.  Rad1 f5
15.  c5 Rf6 16.  Nxe5 f4 17.  Bxf4 Rxf4 18.  Qxf4 dxe5
19.  d6 exf4 20.  Bc4+ Kf8 21.  dxc7 Ke8 22.  e5 Ng5
23.  Rd6 Bxd6 24.  exd6 Bd7 25.  Re1+ Kf8 26.  Re7 Bf5
27.  h4 Nd5 28.  Nxd5 cxd5 29.  Bxd5 Be6 30.  Bxe6 f3
31.  hxg5 b6 32.  Bh3 hxg5 33.  Re5 Rc8 34.  Rf5+ Kg8
35.  Rf7 g4 36.  cxb6 g6 37.  d7 fxg2+ 38.  Kh2 Rf8 39.
Bxg2 Kxf7 40.  Bc6 Ke7 41.  Kg3 Rf3+ 42.  Kh2 Rd3 43.
b7 Rh3+ 44.  Kg2 Re3 45.  d8=Q+ Kf7 46.  b8=Q Rg3+ 47.
Kh2 Rh3+ 48.  Kg1 Rh7 49.  Qe8+ Kf6 50.  Qbd8+ Kf5 51.
Be4+ Kf4 52.  Qf6# 1-0
```

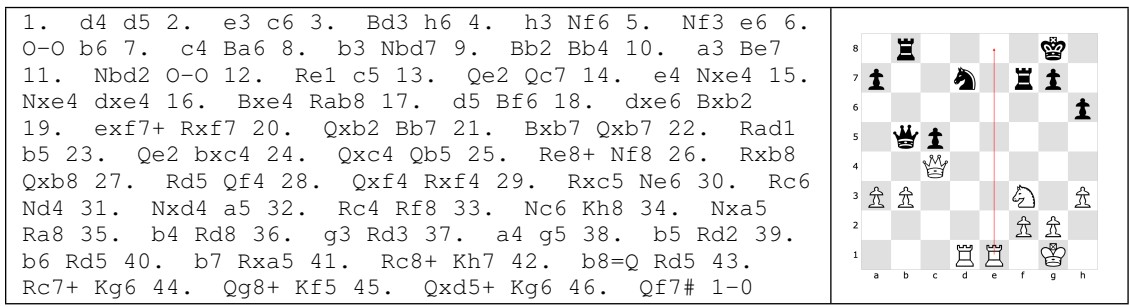

**Game A — SOLIS** ($S$=5, $W$=3) **vs. SF 2350, SOLIS–White. Result: 1–0.** SOLIS sacrifices a knight to gain an advantage in the middlegame.

```
1.  d4 d5 2.  e3 c6 3.  Bd3 h6 4.  h3 Nf6 5.  Nf3 e6 6.
O-O b6 7.  c4 Ba6 8.  b3 Nbd7 9.  Bb2 Bb4 10.  a3 Be7
11.  Nbd2 O-O 12.  Re1 c5 13.  Qe2 Qc7 14.  e4 Nxe4 15.
Nxe4 dxe4 16.  Bxe4 Rab8 17.  d5 Bf6 18.  dxe6 Bxb2
19.  exf7+ Rxf7 20.  Qxb2 Bb7 21.  Bxb7 Qxb7 22.  Rad1
b5 23.  Qe2 bxc4 24.  Qxc4 Qb5 25.  Re8+ Nf8 26.  Rxb8
Qxb8 27.  Rd5 Qf4 28.  Qxf4 Rxf4 29.  Rxc5 Ne6 30.  Rc6
Nd4 31.  Nxd4 a5 32.  Rc4 Rf8 33.  Nc6 Kh8 34.  Nxa5
Ra8 35.  b4 Rd8 36.  g3 Rd3 37.  a4 g5 38.  b5 Rd2 39.
b6 Rd5 40.  b7 Rxa5 41.  Rc8+ Kh7 42.  b8=Q Rd5 43.
Rc7+ Kg6 44.  Qg8+ Kf5 45.  Qxd5+ Kg6 46.  Qf7# 1-0
```

**Game B — SOLIS** ($S$=5, $W$=3) **vs. SF 2300, SOLIS–White. Result: 1–0.** SOLIS sacrifices a rook to win Stockfish's queen, and converts the advantage to a win.

```
1.  e3 Nf6 2.  Nc3 d5 3.  Nf3 c5 4.  d4 e6 5.  g3 a6 6.
a4 Nc6 7.  Ne2 Bd6 8.  Bg2 O-O 9.  h3 Re8 10.  c3 Qc7
11.  Nd2 e5 12.  Nb3 c4 13.  dxe5 Nxe5 14.  Nbd4 Nd3+
15.  Kf1 Nc5 16.  Bf3 Bd7 17.  Kg2 Be5 18.  a5 Rad8
19.  Qc2 g6 20.  Bd2 h5 21.  Be1 h4 22.  gxh4 Kg7 23.
Ng3 Rh8 24.  h5 Rdg8 25.  hxg6 fxg6 26.  Be2 Qd6 27.
Qd1 Qe7 28.  f3 Bd6 29.  Nf1 Rh5 30.  h4 Rgh8 31.  Bf2
R5h7 32.  Qe1 Bb8 33.  Bg3 Ba7 34.  Bf2 Nh5 35.  Ng3
Bb8 36.  f4 Nf6 37.  h5 Rh6 38.  hxg6 Rxh1 39.  Qxh1
Qe8 40.  Ndf5+ Bxf5 41.  Nxf5+ Kxg6 42.  Ne7+ Kf7 43.
e4 Rxh1 44.  Rxh1 Qxe7 45.  Bxc5 Qxe4+ 46.  Bf3 Qg6+
47.  Kf1 Bxf4 48.  b4 cxb3 49.  Rh8 Ke6 50.  Rf8 b2 51.
Rxf6+ Kxf6 52.  Bxd5 b1=R+ 53.  Ke2 Rb2+ 54.  Kf3 Be5
55.  Bc4 Qh5+ 56.  Ke3 Qh3+ 57.  Ke4 Qh7+ 58.  Kf3 Qh3+
59.  Ke4 Qh7+ 60.  Kf3 Qh3+ 61.  Ke4 Qh7+ 62.  Kf3 Qh3+
63.  Ke4 Qh7+ 64.  Kf3 Qh3+ 65.  Ke4 1/2-1/2
```

**Game C — SF 2200 vs. SOLIS** ($S$=5, $W$=3)**, SOLIS–Black. Result: 1/2–1/2.** SOLIS emerges in a winning endgame position, but settles for a draw due to threefold repetition blindness.

```
1.  d4 d5 2.  c4 dxc4 3.  e3 e5 4.  Bxc4 exd4 5.  exd4
Nf6 6.  Nc3 Bd6 7.  Qe2+ Qe7 8.  Be3 O-O 9.  Nf3 c6 10.
O-O Be6 11.  Rfd1 Re8 12.  d5 cxd5 13.  Bxd5 Nc6 14.
a3 Nxd5 15.  Nxd5 Qd8 16.  Qd3 Bg4 17.  Ng5 g6 18.  Ne4
Bxd1 19.  Ndf6+ Kh8 20.  Rxd1 Bxh2+ 21.  Kxh2 Qxd3 22.
Rxd3 Red8 23.  Nd7 Kg7 24.  f3 h6 25.  Rd6 f5 26.  Nec5
Re8 27.  Bd2 Re2 28.  Kh1 Kf7 29.  b3 Ke7 30.  Nxb7 Rc8
31.  Ndc5 Kf7 32.  Kg1 Kg8 33.  Rxg6+ Kf7 34.  Rd6 Ne7
35.  Rd7 Rg8 36.  Nd6+ Kg6 37.  Nc4 Kh5 38.  Ne3 f4 39.
Ne6 fxe3 40.  Nf4+ Kg5 41.  Nxe2 exd2 42.  Kf2 Kf6 43.
Nc3 Ke6 44.  Rd4 Rc8 45.  Ne4 d1=N+ 46.  Rxd1 Rc2+ 47.
Kg1 Nf5 48.  Rd2 Rxd2 49.  Nxd2 Ke5 50.  Kh2 Kf4 51.
b4 Ke3 52.  Ne4 Nd4 53.  a4 Nc2 54.  b5 Nd4 55.  Nd6
Ne6 56.  a5 Kd3 57.  g4 Kd4 58.  b6 axb6 59.  axb6 Nc5
60.  Kg3 Kd5 61.  b7 Na6 62.  Nf7 Kc6 63.  Nxh6 Kxb7
64.  f4 Kc7 65.  Kh4 Kd7 66.  Kh5 Nc5 67.  f5 Kd6 68.
f6 Ne6 69.  Nf7+ Kc6 70.  Kh6 Kd5 71.  g5 Nc5 72.  Kg6
Ne6 73.  Kh5 Kd4 74.  g6 Kc4 75.  Ne5+ Kd5 76.  Ng4 Ke4
77.  Kh6 Kf5 78.  g7 Nxg7 79.  Kxg7 Kxg4 80.  Kf7 Kf5
81.  Ke7 Kg5 82.  f7 Kf4 83.  Kf6 Kf3 84.  Ke5 Kg3 85.
Ke4 Kf2 86.  Kd3 Kg1 87.  Ke3 Kh2 88.  f8=Q Kg3 89.
Qg8+ Kh3 90.  Kf4 Kh2 91.  Kf3 Kh3 92.  Qg3# 1-0
```

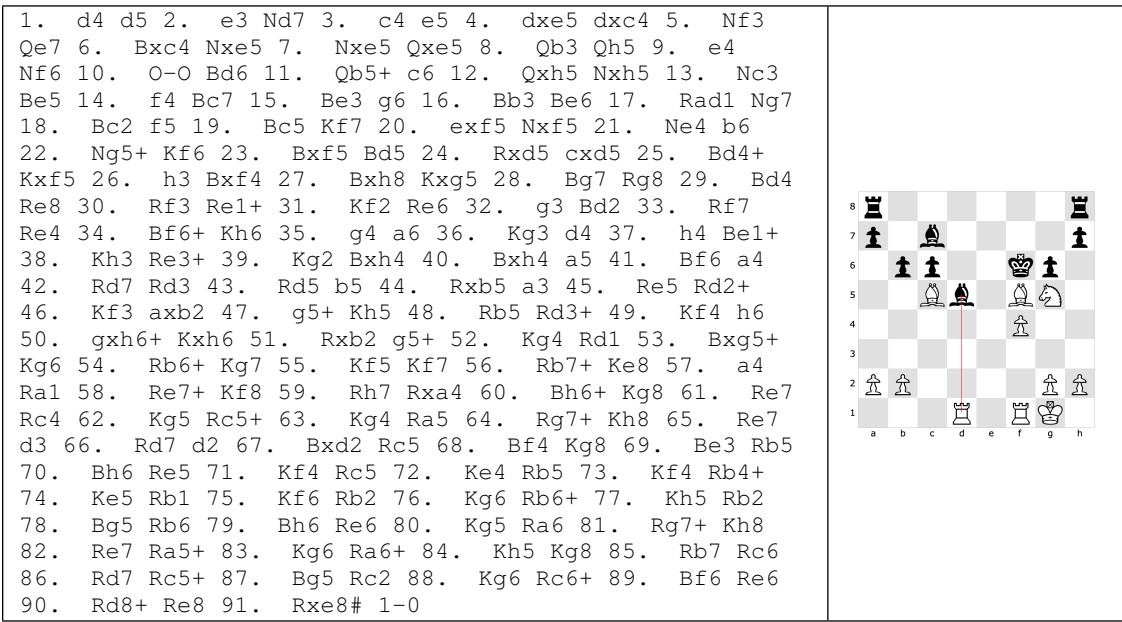

**Game D — SF 2500 vs. SOLIS ($S$=5, $W$=3), SOLIS–Black. Result: 1-0** SOLIS makes a short-sighted piece sacrifice, leading to a long-term material disadvantage.

```
1.  d4 d5 2.  e3 Nd7 3.  c4 e5 4.  dxe5 dxc4 5.  Nf3
Qe7 6.  Bxc4 Nxe5 7.  Nxe5 Qxe5 8.  Qb3 Qh5 9.  e4
Nf6 10.  O-O Bd6 11.  Qb5+ c6 12.  Qxh5 Nxh5 13.  Nc3
Be5 14.  f4 Bc7 15.  Be3 g6 16.  Bb3 Be6 17.  Rad1 Ng7
18.  Bc2 f5 19.  Bc5 Kf7 20.  exf5 Nxf5 21.  Ne4 b6
22.  Ng5+ Kf6 23.  Bxf5 Bd5 24.  Rxd5 cxd5 25.  Bd4+
Kxf5 26.  h3 Bxf4 27.  Bxh8 Kxg5 28.  Bg7 Rg8 29.  Bd4
Re8 30.  Rf3 Re1+ 31.  Kf2 Re6 32.  g3 Bd2 33.  Rf7
Re4 34.  Bf6+ Kh6 35.  g4 a6 36.  Kg3 d4 37.  h4 Be1+
38.  Kh3 Re3+ 39.  Kg2 Bxh4 40.  Bxh4 a5 41.  Bf6 a4
42.  Rd7 Rd3 43.  Rd5 b5 44.  Rxb5 a3 45.  Re5 Rd2+
46.  Kf3 axb2 47.  g5+ Kh5 48.  Rb5 Rd3+ 49.  Kf4 h6
50.  gxh6+ Kxh6 51.  Rxb2 g5+ 52.  Kg4 Rd1 53.  Bxg5+
Kg6 54.  Rb6+ Kg7 55.  Kf5 Kf7 56.  Rb7+ Ke8 57.  a4
Ra1 58.  Re7+ Kf8 59.  Rh7 Rxa4 60.  Bh6+ Kg8 61.  Re7
Rc4 62.  Kg5 Rc5+ 63.  Kg4 Ra5 64.  Rg7+ Kh8 65.  Re7
d3 66.  Rd7 d2 67.  Bxd2 Rc5 68.  Bf4 Kg8 69.  Be3 Rb5
70.  Bh6 Re5 71.  Kf4 Rc5 72.  Ke4 Rb5 73.  Kf4 Rb4+
74.  Ke5 Rb1 75.  Kf6 Rb2 76.  Kg6 Rb6+ 77.  Kh5 Rb2
78.  Bg5 Rb6 79.  Bh6 Re6 80.  Kg5 Ra6 81.  Rg7+ Kh8
82.  Re7 Ra5+ 83.  Kg6 Ra6+ 84.  Kh5 Kg8 85.  Rb7 Rc6
86.  Rd7 Rc5+ 87.  Bg5 Rc2 88.  Kg6 Rc6+ 89.  Bf6 Re6
90.  Rd8+ Re8 91.  Rxe8# 1-0
```

**Game E — SOLIS ($S$=5, $W$=3) vs. Stockfish 2500 , SOLIS–White. Result: 1-0** SOLIS makes an exchange sacrifice for long term compensation, and converts the advantage to a win.

## A.5 EQUAL-DEPTH MATCHES: SOLIS $D$ VS. STOCKFISH $D$

We include several illustrative games where SOLIS and Stockfish search to the same depth $D$. These examples emphasize how SOLIS' latent planning competes head-to-head without a rating cap on Stockfish. The left portion of each table encodes the moves of the game in Portable Game Notation (PGN) format; the right portion of each table displays a critical moment in the game.

```
1.  d4 d5 2.  e3 c5 3.  Nf3 Nf6 4.  Be2 e6 5.  O-O Nc6
6.  b3 cxd4 7.  exd4 Be7 8.  c3 Qc7 9.  h3 Bd7 10.  c4
dxc4 11.  bxc4 Na5 12.  Ne5 Rd8 13.  Bf4 Bd6 14.  c5
Be7 15.  Ng6 Qc8 16.  Nxh8 Bc6 17.  Nc3 Kf8 18.  Rc1
Qa8 19.  Bc7 b5 20.  Bxa5 Bxg2 21.  d5 Bxh3 22.  Qd3
Bf5 23.  Qg3 b4 24.  Rfd1 Bxc5 25.  Qc7 Be7 26.  Nb5
Nxd5 27.  Rxd5 Qxd5 28.  Bxb4 Qd7 29.  Qxd7 Re8 30.
Rc8 g5 31.  Qxe7+ Kg7 32.  Qxf7+ Kh6 33.  Bf8+ Rxf8 34.
Qxf8# 1-0
```

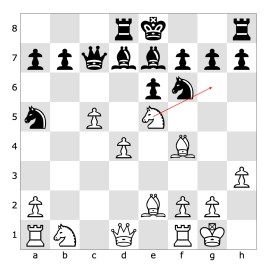

**Game F — SOLIS $S = 5$ vs. Stockfish $S = 5$, SOLIS–White. Result: 1-0** SOLIS sacrifices a knight during the middlegame for a concrete tactical advantage.

```
1.  d4 d5 2.  e3 c5 3.  Nf3 e6 4.  Be2 cxd4 5.  exd4
Bd6 6.  O-O Nf6 7.  Bg5 Nbd7 8.  Nbd2 h6 9.  Bh4 O-O
10.  Re1 a5 11.  Bf1 Bf4 12.  c3 Qb6 13.  Qc2 Qc7 14.
a4 b6 15.  Bb5 Ba6 16.  Bxa6 Rxa6 17.  Qd3 Ra7 18.  Re2
Qc8 19.  Ree1 Nh5 20.  Nf1 Nhf6 21.  N1d2 Nh5 22.  Nf1
Bc7 23.  Qd1 Nhf6 24.  Qd3 Nh5 25.  Qd1 Nhf6 26.  Qd3
Rb7 27.  Qb5 Ra7 28.  N1d2 Nh5 29.  h3 g5 30.  Rac1
gxh4 31.  c4 Nf4 32.  Rc3 Qd8 33.  g3 Nxh3+ 34.  Kg2
Ng5 35.  cxd5 h3+ 36.  Kg1 h2+ 37.  Kxh2 exd5 38.  Rc6
Nf6 39.  Nxg5 hxg5 40.  Kg2 Re8 41.  Rh1 Bd6 42.  Nf3
Kg7 43.  Nxg5 Rc7 44.  Rxd6 Qxd6 45.  Qf1 b5 46.  axb5
Rcc8 47.  Rh4 Rh8 48.  Nf3 Ne4 49.  Rf4 Rh5 50.  Qe2
Qe6 51.  Nh4 Rch8 52.  b6 Nf6 53.  Qf3 a4 54.  b7 Rb8
55.  Rxf6 Qe4 56.  Rxf7+ Kh8 57.  Qxe4 dxe4 58.  Rc7
Rg5 59.  Rc8+ Rg8 60.  Rc7 Rg5 61.  Rc8+ Rg8 62.  Ng6+
Kg7 63.  Ne7 Rf8 64.  Rc7 Rf7 65.  d5 Rbf8 66.  Nf5+
Kg6 67.  Rxf7 Kxf7 68.  Nd6+ Kg6 69.  Nc8 Kg5 70.  b8=Q
Kg4 71.  Qc7 Rf5 72.  Nd6 e3 73.  Qg7+ Rg5 74.  f3+ Kh5
75.  Qh7# 1-0
```

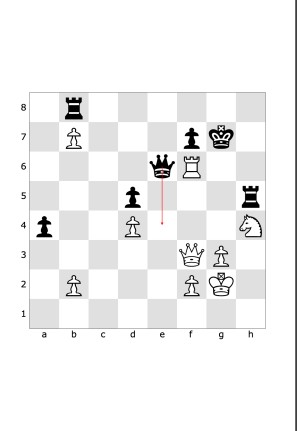

**Game G — SOLIS $S = 5$ vs. Stockfish $S = 5$, SOLIS–White. Result: 1-0** Stockfish steadily gains an advantage, then blunders during the late middlegame. SOLIS capitalizes on this error and wins in the endgame.

```
1.  d4 c6 2.  e4 d5 3.  Nc3 Nf6 4.  e5 Ng8 5.  Bd3 e6
6.  Nf3 c5 7.  dxc5 Bxc5 8.  O-O Nc6 9.  a3 h6 10.  b4
Bb6 11.  Nb5 Bc7 12.  Re1 Bb8 13.  c4 Nge7 14.  Nd6+
Bxd6 15.  exd6 Qxd6 16.  Bb2 O-O 17.  b5 dxc4 18.  Bh7+
Kh8 19.  Qxd6 f6 20.  Be4 c3 21.  Bxc3 e5 22.  bxc6 Bf5
23.  Qxe7 Bxe4 24.  Rxe4 bxc6 25.  Nxe5 Rfe8 26.  Nf7+
Kg8 27.  Nxh6+ Kh7 28.  Rae1 Rad8 29.  Ba5 Rd4 30.  g4
Rxe4 31.  Rxe4 Rb8 32.  Qxa7 Rd8 33.  Rf4 Rd3 34.  g5
fxg5 35.  Rd4 Rxa3 36.  Kf1 Rb3 37.  Rb4 Rc3 38.  h4
gxh4 39.  Rd4 h3 40.  Qe7 h2 41.  Qe4+ Kxh6 42.  Rd6+
Kg5 43.  Rg6+ Kh5 44.  Qg4# 1-0
```

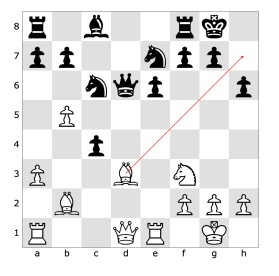

**Game H — SOLIS $S = 3$ vs. Stockfish $S = 3$, SOLIS–White. Result: 1-0** SOLIS sacrifices a bishop for a discovered attack on Stockfish's queen, and converts the advantage to a win.

```
1.  d4 d5 2.  e3 c6 3.  c4 e6 4.  Nf3 Nf6 5.  Be2 c5
6.  O-O Nc6 7.  b3 a6 8.  Bb2 Be7 9.  a3 cxd4 10.  exd4
O-O 11.  c5 b6 12.  b4 bxc5 13.  dxc5 Bb7 14.  Nbd2 a5
15.  b5 Bxc5 16.  Rc1 Qb6 17.  bxc6 Ng4 18.  Rxc5 Ba6
19.  Rc2 Nxf2 20.  Rxf2 Bxe2 21.  Qxe2 a4 22.  c7 Rac8
23.  Bd4 Qb5 24.  Qxb5 Rfe8 25.  Qb8 h6 26.  Ne5 g5 27.
h4 Rf8 28.  Ne4 dxe4 29.  Rxf7 Rxb8 30.  Rxf8+ Rxf8 31.
hxg5 Rc8 32.  gxh6 e3 33.  Bxe3 Kh7 34.  Rc4 Kg8 35.
Kh2 Kh7 36.  g4 Rg8 37.  g5 Ra8 38.  Rg4 Kh8 39.  Ba7
Rf8 40.  Bb8 Rf2+ 41.  Kg1 Rc2 42.  Rc4 Rb2 43.  Ba7
Rb1+ 44.  Kg2 Kh7 45.  Bf2 Rb2 46.  g6+ Kxh6 47.  Kh2
Rxf2+ 48.  Kg3 Rg2+ 49.  Kxg2 Kg7 50.  Rc1 Kf6 51.  Kh2
Kg7 52.  Nd7 e5 53.  Nf6 Kxf6 54.  Rg1 Kg7 55.  Rf1 Kh6
56.  Rg1 Kg7 57.  Rf1 Kxg6 58.  Rc1 Kg7 59.  Rc4 Kf7
60.  Kg2 Ke7 61.  Kf2 Kd6 62.  Ke1 Kd5 63.  Rc1 Ke4 64.
Ke2 Kd4 65.  Rc2 e4 66.  Rc1 Kd5 67.  Ke1 Ke5 68.  Rc4
Kd5 69.  Rc2 Kd6 70.  Rc1 Ke7 71.  Rc4 Kf7 72.  Rc5 e3
73.  Kf1 Ke7 74.  Rc4 Kf7 75.  Kg2 e2 76.  Kf2 Ke7 77.
Ke1 Kf7 78.  Rc5 Kg7 79.  Rc2 Kf6 80.  Rc4 Kf7 81.  Rc5
Kg6 82.  Kd2 Kf7 83.  Ke1 Ke7 84.  Rc4 Kf7 85.  Rc5 Kg6
86.  Rc4 Kf7 87.  Rc5 1/2-1/2
```

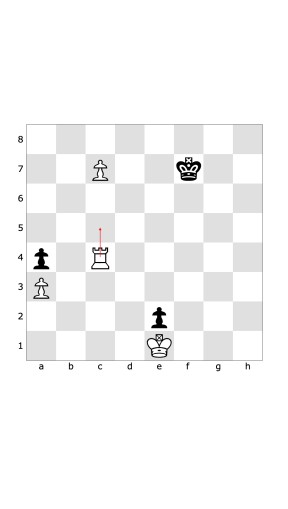

**Game I — SOLIS** $S = 3$ **vs. Stockfish** $S = 3$**, SOLIS–White. Result: 1/2-1/2** SOLIS struggles to plan in a position where every trajectory is winning.

## A.6  SOLIS vs. Stockfish Results

To support reproducibility of our Elo estimates, we provide the full match outcomes between SOLIS and Stockfish, where SOLIS searches at depths $S \in \{2, 3, 4, 5\}$. Each table is indexed by Stockfish's Elo cap (rows) and SOLIS search width $W$ (columns). Entries are given as win–loss–draw counts from the matchups, with total match points in parentheses. Separate tables are reported for the Base and Mini models, and for both unanchored and anchored scoring.

**Base: Depth 2**

| SF Elo | W=1 | W=2 | W=3 | W=5 |
|---|---|---|---|---|
| 1500 | 32–46–22 (55) | 41–47–12 (64) | – | – |
| 1550 | 39–38–23 (58) | 37–50–13 (62) | – | – |
| 1600 | 32–44–24 (54) | 33–51–16 (58) | – | – |
| 1650 | 33–41–26 (54) | 29–46–25 (52) | – | – |
| 1700 | 34–38–28 (53) | 22–53–25 (48) | – | – |
| 1750 | 37–35–28 (54) | 30–49–21 (54) | – | – |
| 1800 | 29–42–29 (50) | 36–42–22 (57) | – | – |
| 1850 | – | – | – | – |
| 1900 | – | – | – | – |
| 1950 | – | – | – | – |
| 2000 | – | – | 34–40–26 (54) | 43–31–26 (58) |
| 2050 | – | – | 46–38–16 (65) | 36–38–26 (55) |
| 2100 | – | – | 34–45–21 (56) | 34–31–35 (50) |
| 2150 | – | – | 46–31–23 (62) | 30–43–27 (52) |
| 2200 | – | – | 31–35–34 (48) | 35–28–37 (49) |
| 2250 | – | – | 34–34–32 (51) | 40–29–31 (54) |
| 2300 | – | – | 32–32–36 (48) | 37–34–29 (54) |
| 2350 | – | – | – | – |
| 2400 | – | – | – | – |
| 2450 | – | – | – | – |
| 2500 | – | – | – | – |
| 2550 | – | – | – | – |
| 2600 | – | – | – | – |
| 2650 | – | – | – | – |
| 2700 | – | – | – | – |

**Base: Depth 3**

| SF Elo | W=1 | W=2 | W=3 | W=5 |
|---|---|---|---|---|
| 1500 | – | – | – | – |
| 1550 | – | – | – | – |
| 1600 | – | – | – | – |
| 1650 | – | – | – | – |
| 1700 | – | – | – | – |
| 1750 | – | – | – | – |
| 1800 | 39–29–32 (54) | 37–49–14 (62) | – | – |
| 1850 | 35–32–33 (51) | 30–50–20 (55) | – | – |
| 1900 | 37–30–33 (52) | 40–41–19 (60) | – | – |
| 1950 | 31–28–41 (45) | 37–44–19 (59) | – | – |
| 2000 | 41–27–32 (54) | 33–45–22 (56) | – | – |
| 2050 | 39–27–34 (52) | 31–41–28 (52) | – | – |
| 2100 | 41–27–32 (54) | 36–38–26 (55) | – | – |
| 2150 | – | – | – | – |
| 2200 | – | – | 33–42–25 (54) | 30–41–29 (50) |
| 2250 | – | – | 35–41–24 (56) | 33–43–24 (54) |
| 2300 | – | – | 35–33–32 (52) | 36–34–30 (53) |
| 2350 | – | – | 37–37–26 (56) | 29–37–34 (48) |
| 2400 | – | – | 34–36–30 (52) | 37–38–25 (56) |
| 2450 | – | – | 37–28–35 (51) | 43–29–28 (58) |
| 2500 | – | – | 41–30–29 (56) | 34–30–36 (49) |
| 2550 | – | – | – | – |
| 2600 | – | – | – | – |
| 2650 | – | – | – | – |
| 2700 | – | – | – | – |

**Base: Depth 4**

| SF Elo | W=1 | W=2 | W=3 | W=5 |
|---|---|---|---|---|
| 1500 | – | – | – | – |
| 1550 | – | – | – | – |
| 1600 | – | – | – | – |
| 1650 | – | – | – | – |
| 1700 | – | – | – | – |
| 1750 | – | – | – | – |
| 1800 | – | – | – | – |
| 1850 | – | – | – | – |
| 1900 | – | – | – | – |
| 1950 | 37–26–37 (50) | 41–42–17 (62) | – | – |
| 2000 | 43–20–37 (53) | 31–43–26 (52) | – | – |
| 2050 | 38–27–35 (52) | 40–38–22 (59) | – | – |
| 2100 | 42–25–33 (54) | 32–48–20 (56) | – | – |
| 2150 | 36–23–41 (48) | 31–47–22 (54) | – | – |
| 2200 | 46–13–41 (52) | 39–39–22 (58) | – | – |
| 2250 | – | – | – | – |
| 2300 | – | – | – | – |
| 2350 | – | – | – | – |
| 2400 | – | – | – | – |
| 2450 | – | – | 40–39–21 (60) | 39–38–23 (58) |
| 2500 | – | – | 38–38–24 (57) | 41–40–19 (61) |
| 2550 | – | – | 44–20–36 (54) | 42–29–29 (56) |
| 2600 | – | – | 46–17–37 (54) | 37–23–40 (48) |
| 2650 | – | – | 41–18–41 (50) | 44–22–34 (55) |
| 2700 | – | – | 45–16–39 (53) | 42–22–36 (53) |

**Base: Depth 5**

| SF Elo | W=1 | W=2 | W=3 | W=5 |
|---|---|---|---|---|
| 1500 | – | – | – | – |
| 1550 | – | – | – | – |
| 1600 | – | – | – | – |
| 1650 | – | – | – | – |
| 1700 | – | – | – | – |
| 1750 | – | – | – | – |
| 1800 | – | – | – | – |
| 1850 | – | – | – | – |
| 1900 | – | – | – | – |
| 1950 | 36–34–30 (53) | 38–41–21 (58) | – | – |
| 2000 | 31–28–41 (45) | 27–52–21 (53) | – | – |
| 2050 | 41–24–35 (53) | 36–37–27 (54) | – | – |
| 2100 | 43–23–34 (54) | 36–39–25 (56) | – | – |
| 2150 | 42–23–35 (54) | 27–45–28 (50) | – | – |
| 2200 | 42–21–37 (52) | 23–51–26 (48) | – | – |
| 2250 | – | – | – | – |
| 2300 | – | – | – | – |
| 2350 | – | – | – | – |
| 2400 | – | – | – | – |
| 2450 | – | – | 25–46–29 (48) | 43–30–27 (58) |
| 2500 | – | – | 40–36–24 (58) | 34–44–22 (56) |
| 2550 | – | – | 40–31–29 (56) | 38–38–24 (57) |
| 2600 | – | – | 46–25–29 (58) | 33–37–30 (52) |
| 2650 | – | – | 47–17–36 (56) | 30–35–35 (48) |
| 2700 | – | – | 38–25–37 (50) | 45–21–34 (56) |

Figure 6: Base model vs. Stockfish: per-depth blocks with unanchored planning. W–D–L (score) out of 100.

**Mini: Depth 2**

| SF Elo | W=1 | W=2 | W=3 | W=5 |
|---|---|---|---|---|
| 1350 | 32–32–36 (48) | 27–45–28 (50) | – | – |
| 1400 | 30–41–29 (50) | 33–27–40 (46) | – | – |
| 1450 | 29–44–27 (51) | 33–40–27 (53) | – | – |
| 1500 | 31–38–31 (50) | 31–39–30 (50) | – | – |
| 1550 | 38–35–27 (56) | 23–50–27 (48) | – | – |
| 1600 | 34–30–36 (49) | 29–37–34 (48) | – | – |
| 1650 | 24–45–31 (46) | 28–36–36 (46) | – | – |
| 1700 | – | – | – | – |
| 1750 | – | – | – | – |
| 1800 | – | – | 37–31–32 (52) | 37–36–27 (55) |
| 1850 | – | – | 34–24–42 (46) | 34–31–35 (50) |
| 1900 | – | – | 40–28–32 (54) | 40–29–31 (54) |
| 1950 | – | – | 39–29–32 (54) | 39–29–32 (54) |
| 2000 | – | – | 38–34–28 (55) | 42–28–30 (56) |
| 2050 | – | – | 41–27–32 (54) | 40–26–34 (53) |
| 2100 | – | – | 35–30–35 (50) | 36–26–38 (49) |
| 2150 | – | – | – | – |
| 2200 | – | – | – | – |
| 2250 | – | – | – | – |
| 2300 | – | – | – | – |
| 2350 | – | – | – | – |
| 2400 | – | – | – | – |
| 2500 | – | – | – | – |

**Mini: Depth 3**

| SF Elo | W=1 | W=2 | W=3 | W=5 |
|---|---|---|---|---|
| 1350 | – | – | – | – |
| 1400 | – | – | – | – |
| 1450 | – | – | – | – |
| 1500 | 29–38–33 (48) | 28–43–29 (50) | – | – |
| 1550 | 41–36–23 (59) | 36–37–27 (54) | – | – |
| 1600 | 37–30–33 (52) | 37–30–33 (52) | – | – |
| 1650 | 32–38–30 (51) | 30–40–30 (50) | – | – |
| 1700 | 28–37–35 (46) | 21–43–36 (42) | – | – |
| 1750 | 36–33–31 (52) | 28–40–32 (48) | – | – |
| 1800 | 39–24–37 (51) | 29–35–36 (46) | – | – |
| 1850 | – | – | – | – |
| 1900 | – | – | – | – |
| 1950 | – | – | – | – |
| 2000 | – | – | 34–35–31 (52) | 38–21–41 (48) |
| 2050 | – | – | 38–34–28 (55) | 34–33–33 (50) |
| 2100 | – | – | 33–34–33 (50) | 34–29–37 (48) |
| 2150 | – | – | 31–35–34 (48) | 38–25–37 (50) |
| 2200 | – | – | 38–18–44 (47) | 31–26–43 (44) |
| 2250 | – | – | 36–23–41 (48) | 45–26–29 (58) |
| 2300 | – | – | 40–22–38 (51) | 34–26–40 (47) |
| 2350 | – | – | – | – |
| 2400 | – | – | – | – |
| 2500 | – | – | – | – |

**Mini: Depth 4**

| SF Elo | W=1 | W=2 | W=3 | W=5 |
|---|---|---|---|---|
| 1350 | – | – | – | – |
| 1400 | – | – | – | – |
| 1450 | – | – | – | – |
| 1500 | – | – | – | – |
| 1550 | – | – | – | – |
| 1600 | 36–42–22 (57) | 35–31–34 (50) | – | – |
| 1650 | 29–40–31 (49) | 36–33–31 (52) | – | – |
| 1700 | 33–34–33 (50) | 36–32–32 (52) | – | – |
| 1750 | 43–26–31 (56) | 28–38–34 (47) | – | – |
| 1800 | 38–24–38 (50) | 30–37–33 (48) | – | – |
| 1850 | 49–23–28 (60) | 29–33–38 (46) | – | – |
| 1900 | – | – | – | – |
| 1950 | – | – | – | – |
| 2000 | – | – | – | – |
| 2050 | – | – | – | – |
| 2100 | – | – | – | – |
| 2150 | – | – | 31–31–38 (46) | 44–21–35 (54) |
| 2200 | – | – | 35–29–36 (50) | 48–27–25 (62) |
| 2250 | – | – | 37–31–32 (52) | 37–28–35 (51) |
| 2300 | – | – | 27–37–36 (46) | 36–34–30 (53) |
| 2350 | – | – | 22–42–36 (43) | 37–35–28 (54) |
| 2400 | – | – | 36–29–35 (50) | 37–26–37 (50) |
| 2500 | – | – | – | – |

**Mini: Depth 5**

| SF Elo | W=1 | W=2 | W=3 | W=5 |
|---|---|---|---|---|
| 1350 | – | – | – | – |
| 1400 | – | – | – | – |
| 1450 | – | – | – | – |
| 1500 | – | – | – | – |
| 1550 | – | – | – | – |
| 1600 | – | – | – | – |
| 1650 | – | – | – | – |
| 1700 | 39–30–31 (54) | 34–34–32 (51) | – | – |
| 1750 | 37–29–34 (52) | 32–30–38 (47) | – | – |
| 1800 | 42–20–38 (52) | 33–31–36 (48) | – | – |
| 1850 | 36–24–40 (48) | 43–28–29 (57) | – | – |
| 1900 | 45–26–29 (58) | 46–31–23 (62) | – | – |
| 1950 | 40–22–38 (51) | 37–35–28 (54) | – | – |
| 2000 | – | – | – | – |
| 2050 | – | – | – | – |
| 2100 | – | – | – | – |
| 2150 | – | – | – | – |
| 2200 | – | – | 27–37–36 (46) | 33–38–29 (52) |
| 2250 | – | – | 30–33–37 (46) | 38–31–31 (54) |
| 2300 | – | – | 38–37–25 (56) | 33–39–28 (52) |
| 2350 | – | – | 32–29–39 (46) | 29–36–35 (47) |
| 2400 | – | – | 30–26–44 (43) | 30–32–38 (46) |
| 2500 | – | – | 26–34–40 (43) | 43–25–32 (56) |

Figure 7: Mini model vs. Stockfish: per-depth blocks with unanchored planning. W–D–L (score) out of 100.

**Base: Depth 2**

| SF Elo | W=1 | W=2 | W=3 | W=5 |
|---|---|---|---|---|
| 1500 | 32–46–22 (55) | 41–47–12 (64) | – | – |
| 1550 | 39–38–23 (58) | 37–50–13 (62) | – | – |
| 1600 | 32–44–24 (54) | 33–51–16 (58) | – | – |
| 1650 | 33–41–26 (54) | 29–46–25 (52) | – | – |
| 1700 | 34–38–28 (53) | 22–53–25 (48) | – | – |
| 1750 | 37–35–28 (54) | 30–49–21 (54) | – | – |
| 1800 | 29–42–29 (50) | 36–42–22 (57) | – | – |
| 1850 | – | – | – | – |
| 1900 | – | – | – | – |
| 1950 | – | – | – | – |
| 2000 | – | – | 34–40–26 (54) | 43–31–26 (58) |
| 2050 | – | – | 46–38–16 (65) | 36–38–26 (55) |
| 2100 | – | – | 34–45–21 (56) | 34–31–35 (50) |
| 2150 | – | – | 46–31–23 (62) | 30–43–27 (52) |
| 2200 | – | – | 31–35–34 (48) | 35–28–37 (49) |
| 2250 | – | – | 34–34–32 (51) | 40–29–31 (54) |
| 2300 | – | – | 32–32–36 (48) | 37–34–29 (54) |
| 2350 | – | – | – | – |
| 2400 | – | – | – | – |
| 2450 | – | – | – | – |
| 2500 | – | – | – | – |
| 2550 | – | – | – | – |
| 2600 | – | – | – | – |
| 2650 | – | – | – | – |
| 2700 | – | – | – | – |

**Base: Depth 3**

| SF Elo | W=1 | W=2 | W=3 | W=5 |
|---|---|---|---|---|
| 1500 | – | – | – | – |
| 1550 | – | – | – | – |
| 1600 | – | – | – | – |
| 1650 | – | – | – | – |
| 1700 | – | – | – | – |
| 1750 | – | – | – | – |
| 1800 | 39–29–32 (54) | 37–49–14 (62) | – | – |
| 1850 | 35–32–33 (51) | 30–50–20 (55) | – | – |
| 1900 | 37–30–33 (52) | 40–41–19 (60) | – | – |
| 1950 | 31–28–41 (45) | 37–44–19 (59) | – | – |
| 2000 | 41–27–32 (54) | 33–45–22 (56) | – | – |
| 2050 | 39–27–34 (52) | 31–41–28 (52) | – | – |
| 2100 | 41–27–32 (54) | 36–38–26 (55) | – | – |
| 2150 | – | – | – | – |
| 2200 | – | – | 33–42–25 (54) | 30–41–29 (50) |
| 2250 | – | – | 35–41–24 (56) | 33–43–24 (54) |
| 2300 | – | – | 35–33–32 (52) | 36–34–30 (53) |
| 2350 | – | – | 37–37–26 (56) | 29–37–34 (48) |
| 2400 | – | – | 34–36–30 (52) | 37–38–25 (56) |
| 2450 | – | – | 37–28–35 (51) | 43–29–28 (58) |
| 2500 | – | – | 41–30–29 (56) | 34–30–36 (49) |
| 2550 | – | – | – | – |
| 2600 | – | – | – | – |
| 2650 | – | – | – | – |
| 2700 | – | – | – | – |

**Base: Depth 4**

| SF Elo | W=1 | W=2 | W=3 | W=5 |
|---|---|---|---|---|
| 1500 | – | – | – | – |
| 1550 | – | – | – | – |
| 1600 | – | – | – | – |
| 1650 | – | – | – | – |
| 1700 | – | – | – | – |
| 1750 | – | – | – | – |
| 1800 | – | – | – | – |
| 1850 | – | – | – | – |
| 1900 | – | – | – | – |
| 1950 | 37–26–37 (50) | 41–42–17 (62) | – | – |
| 2000 | 43–20–37 (53) | 31–43–26 (52) | – | – |
| 2050 | 38–27–35 (52) | 40–38–22 (59) | – | – |
| 2100 | 42–25–33 (54) | 32–48–20 (56) | – | – |
| 2150 | 36–23–41 (48) | 31–47–22 (54) | – | – |
| 2200 | 46–13–41 (52) | 39–39–22 (58) | – | – |
| 2250 | – | – | – | – |
| 2300 | – | – | – | – |
| 2350 | – | – | – | – |
| 2400 | – | – | – | – |
| 2450 | – | – | 40–39–21 (60) | 39–38–23 (58) |
| 2500 | – | – | 38–38–24 (57) | 41–40–19 (61) |
| 2550 | – | – | 44–20–36 (54) | 42–29–29 (56) |
| 2600 | – | – | 46–17–37 (54) | 37–23–40 (48) |
| 2650 | – | – | 41–18–41 (50) | 44–22–34 (55) |
| 2700 | – | – | 45–16–39 (53) | 42–22–36 (53) |

**Base: Depth 5**

| SF Elo | W=1 | W=2 | W=3 | W=5 |
|---|---|---|---|---|
| 1500 | – | – | – | – |
| 1550 | – | – | – | – |
| 1600 | – | – | – | – |
| 1650 | – | – | – | – |
| 1700 | – | – | – | – |
| 1750 | – | – | – | – |
| 1800 | – | – | – | – |
| 1850 | – | – | – | – |
| 1900 | – | – | – | – |
| 1950 | 36–34–30 (53) | 38–41–21 (58) | – | – |
| 2000 | 31–28–41 (45) | 27–52–21 (53) | – | – |
| 2050 | 41–24–35 (53) | 36–37–27 (54) | – | – |
| 2100 | 43–23–34 (54) | 36–39–25 (56) | – | – |
| 2150 | 42–23–35 (54) | 27–45–28 (50) | – | – |
| 2200 | 42–21–37 (52) | 23–51–26 (48) | – | – |
| 2250 | – | – | – | – |
| 2300 | – | – | – | – |
| 2350 | – | – | – | – |
| 2400 | – | – | – | – |
| 2450 | – | – | 25–46–29 (48) | 43–30–27 (58) |
| 2500 | – | – | 40–36–24 (58) | 34–44–22 (56) |
| 2550 | – | – | 40–31–29 (56) | 38–38–24 (57) |
| 2600 | – | – | 46–25–29 (58) | 33–37–30 (52) |
| 2650 | – | – | 47–17–36 (56) | 30–35–35 (48) |
| 2700 | – | – | 38–25–37 (50) | 45–21–34 (56) |

Figure 8: Base model vs. Stockfish: per-depth blocks with anchored planning. W–D–L (score) out of 100.

**Mini: Depth 2**

| SF Elo | W=1 | W=2 | W=3 | W=5 |
|---|---|---|---|---|
| 1350 | 36–32–32 (52) | 39–30–31 (54) | – | – |
| 1400 | 23–49–28 (48) | 30–32–38 (46) | – | – |
| 1450 | 33–37–30 (52) | 24–47–29 (48) | – | – |
| 1500 | 34–38–28 (53) | 25–40–35 (45) | – | – |
| 1550 | 28–38–34 (47) | 40–39–21 (60) | – | – |
| 1600 | 23–47–30 (46) | 28–42–30 (49) | – | – |
| 1650 | 35–29–36 (50) | 38–37–25 (56) | – | – |
| 1700 | – | – | – | – |
| 1750 | – | – | – | – |
| 1800 | – | – | 35–37–28 (54) | 40–28–32 (54) |
| 1850 | – | – | 32–32–36 (48) | 44–29–27 (58) |
| 1900 | – | – | 37–30–33 (52) | 45–21–34 (56) |
| 1950 | – | – | 43–30–27 (58) | 36–29–35 (50) |
| 2000 | – | – | 35–31–34 (50) | 43–27–30 (56) |
| 2050 | – | – | 37–29–34 (52) | 45–27–28 (58) |
| 2100 | – | – | 36–27–37 (50) | 42–34–24 (59) |
| 2150 | – | – | – | – |
| 2200 | – | – | – | – |
| 2250 | – | – | – | – |
| 2300 | – | – | – | – |
| 2350 | – | – | – | – |
| 2400 | – | – | – | – |
| 2500 | – | – | – | – |

**Mini: Depth 3**

| SF Elo | W=1 | W=2 | W=3 | W=5 |
|---|---|---|---|---|
| 1350 | – | – | – | – |
| 1400 | – | – | – | – |
| 1450 | – | – | – | – |
| 1500 | 29–40–31 (49) | 35–33–32 (52) | – | – |
| 1550 | 34–35–31 (52) | 37–29–34 (52) | – | – |
| 1600 | 32–36–32 (50) | 32–39–29 (52) | – | – |
| 1650 | 34–30–36 (49) | 37–33–30 (54) | – | – |
| 1700 | 36–30–34 (51) | 34–35–31 (52) | – | – |
| 1750 | 34–25–41 (46) | 34–30–36 (49) | – | – |
| 1800 | 32–29–39 (46) | 32–31–37 (48) | – | – |
| 1850 | – | – | – | – |
| 1900 | – | – | – | – |
| 1950 | – | – | – | – |
| 2000 | – | – | 46–32–22 (62) | 38–27–35 (52) |
| 2050 | – | – | 37–27–36 (50) | 39–26–35 (52) |
| 2100 | – | – | 29–34–37 (46) | 48–17–35 (56) |
| 2150 | – | – | 37–33–30 (54) | 39–25–36 (52) |
| 2200 | – | – | 37–31–32 (52) | 41–23–36 (52) |
| 2250 | – | – | 36–23–41 (48) | 44–20–36 (54) |
| 2300 | – | – | 30–43–27 (52) | 38–26–36 (51) |
| 2350 | – | – | – | – |
| 2400 | – | – | – | – |
| 2500 | – | – | – | – |

**Mini: Depth 4**

| SF Elo | W=1 | W=2 | W=3 | W=5 |
|---|---|---|---|---|
| 1350 | – | – | – | – |
| 1400 | – | – | – | – |
| 1450 | – | – | – | – |
| 1500 | – | – | – | – |
| 1550 | – | – | – | – |
| 1600 | 26–45–29 (48) | 31–33–36 (48) | – | – |
| 1650 | 37–39–24 (56) | 28–32–40 (44) | – | – |
| 1700 | 23–39–38 (42) | 29–41–30 (50) | – | – |
| 1750 | 36–29–35 (50) | 38–30–32 (53) | – | – |
| 1800 | 35–24–41 (47) | 33–36–31 (51) | – | – |
| 1850 | 41–21–38 (52) | 30–28–42 (44) | – | – |
| 1900 | – | – | – | – |
| 1950 | – | – | – | – |
| 2000 | – | – | – | – |
| 2050 | – | – | – | – |
| 2100 | – | – | – | – |
| 2150 | – | – | 39–23–38 (50) | 32–30–38 (47) |
| 2200 | – | – | 33–40–27 (53) | 38–25–37 (50) |
| 2250 | – | – | 36–31–33 (52) | 30–27–43 (44) |
| 2300 | – | – | 32–33–35 (48) | 37–28–35 (51) |
| 2350 | – | – | 35–34–31 (52) | 40–25–35 (52) |
| 2400 | – | – | 28–26–46 (41) | 26–38–36 (45) |
| 2500 | – | – | – | – |

**Mini: Depth 5**

| SF Elo | W=1 | W=2 | W=3 | W=5 |
|---|---|---|---|---|
| 1350 | – | – | – | – |
| 1400 | – | – | – | – |
| 1450 | – | – | – | – |
| 1500 | – | – | – | – |
| 1550 | – | – | – | – |
| 1600 | – | – | – | – |
| 1650 | – | – | – | – |
| 1700 | 37–36–27 (55) | 34–29–37 (48) | – | – |
| 1750 | 42–23–35 (54) | 37–33–30 (54) | – | – |
| 1800 | 36–28–36 (50) | 31–35–34 (48) | – | – |
| 1850 | 38–21–41 (48) | 33–32–35 (49) | – | – |
| 1900 | 37–27–36 (50) | 37–37–26 (56) | – | – |
| 1950 | 43–18–39 (52) | 38–24–38 (50) | – | – |
| 2000 | – | – | – | – |
| 2050 | – | – | – | – |
| 2100 | – | – | – | – |
| 2150 | – | – | – | – |
| 2200 | – | – | 30–40–30 (50) | 32–34–34 (49) |
| 2250 | – | – | 29–43–28 (50) | 49–26–25 (62) |
| 2300 | – | – | 31–30–39 (46) | 30–35–35 (48) |
| 2350 | – | – | 26–32–42 (42) | 40–34–26 (57) |
| 2400 | – | – | 39–32–29 (55) | 31–32–37 (47) |
| 2500 | – | – | 30–29–41 (44) | 28–41–31 (48) |

Figure 9: Mini model vs. Stockfish: per-depth blocks with anchored planning. W–D–L (score) out of 100.

