# OpenReview forum: "Latent Planning via Embedding Arithmetic: A Contrastive Approach to Strategic Reasoning"
_ICLR.cc/2026/Conference — Submitted to ICLR 2026_

### Official Review · Reviewer_gbUS · 2025-10-27

**Soundness:** 2
**Presentation:** 2
**Contribution:** 2
**Rating:** 2
**Confidence:** 3

**Summary:**

the paper proposes SOLIS, a method that performs planning in a learned embedding space instead of "relying on explicit policies or value networks". The embedding is trained with supervised contrastive learning so that "outcome similarity is captured by proximity", and a "single global advantage vector orients the space from losing to winning regions". In chess, SOLIS uses this structure to guide shallow search and shows competitive results under limited conditions.

**Strengths:**

The scope and the benchmark considered are interesting

**Weaknesses:**

The paper lacks technical details on the approach both
- from a theoretical or at least technical perspective as well as
- from the experiments.

**Questions:**

-  what does the following mean "treating planning as traversal through a high-dimensional embedding space."
- What is the planning procedure in details beyond general statements such as "At inference we plan directly in the learned space by ranking legal continuations using a single global advantage direction.". How does it select moves to expand based on the branching factor of 3 mentioned in the paper.
- Does SOLIS maintain similar relative performance if applied to non-chess planning tasks?
- How do the findings inform the broader claim that planning can be done in latent space without explicit dynamics models?

---

### Official Review · Reviewer_42X5 · 2025-11-06

**Soundness:** 3
**Presentation:** 3
**Contribution:** 2
**Rating:** 4
**Confidence:** 4

**Summary:**

This manuscript proposes to learn a latent space via a supervised contrastive learning encoder that can then be used for planning. The latent space is navigated through the use of an advantage vector. Results are shown on chess games where different depths and widths of search improve performance.

**Strengths:**

This paper is extremely well written. It is clear from the paper what it's contributions are, what the method is, and what problem the paper is attempting to solve. The method is simple, and likely can transfer to other domains where search can be used.

**Weaknesses:**

1. I have concerns about the lack of included baselines. One of the papers that I am fairly familiar with in this area was not mentioned at all [1], while the mention of AlphaZero also gave me the expectation that it would be compared in the work as well as it is also one of the main algorithms in this space. A comparison to the various Dreamer versions could also be very useful here.

2. I think Table 2 is missing information about the configuration on how these results were generated. How many random seeds are used to generate the results? How meaningful is a difference between these scores? Are the results statistically significant?

Minor Comments
Line 46: Is 2500+ Elo good? Contextualize it a bit for the reader.








Citations
1. https://proceedings.mlr.press/v139/ozair21a/ozair21a-supp.pdf

**Questions:**

1. What is the data distribution look like in ChessBench?
2. How could different search methods (i.e. MCTS) be integrated into the SOLIS method?
3. The embedding guided search is a fairly simple and general method, but seems to leave some improvement due to "randomly" traversing the search space. How could search methods leverage the knowledge captured in the latent embedding space to improve search?

---

### Official Review · Reviewer_2c4V · 2025-11-08

**Soundness:** 3
**Presentation:** 4
**Contribution:** 3
**Rating:** 6
**Confidence:** 3

**Summary:**

The paper introduces SOLIS, a transformer-based encoder trained with supervised contrastive learning to embed chess positions into a latent space aligned with game outcomes. In this space, positions that lead to similar outcomes are embedded close to one another, and a single advantage vector is used to distinguish between winning and losing positions. Planning is then performed by ranking legal moves based on their alignment with this vector, combined with a lightweight min–max search over the top few candidates.

**Strengths:**

* The paper clearly articulates its main ideas and presents them in an accessible way. It effectively demonstrates how to construct an evaluation-aligned latent space using contrastive learning, extract a single advantage vector, and use this direction to guide a compact search procedure.
* The two scoring variants, anchored and unanchored, are intuitive and well explained.
* The latent space visualizations are insightful and visually engaging, helping to illustrate the internal structure of the learned representations.
* The ablation studies across search depth and width are well executed and clearly demonstrate how performance scales with model size and scoring variant.
* The paper explicitly acknowledges several limitations, such as the non-Markovian nature of the input representation and the inability to handle repetitions or prefer shorter forced wins, and illustrates these limitations with concrete examples in the appendix.

**Weaknesses:**

The paper has two main issues that needs clarification and refinement:
* The construction of the single global “advantage vector” is underspecified and potentially fragile
* Evaluation framing and the “2500+ Elo” claim are somewhat misleading

*Issue 1*:
The central idea of the paper is that a single vector in the embedding space captures the direction from “losing” to “winning” positions. However, the process of creating a single advantage vector by averaging the embedding across extreme positions can have potentially few issues in it. First, how large and diverse are these extreme sets? If they mostly contain trivial mates or blunders, the resulting direction could overfit to those patterns. Second, how sensitive is this vector to hyperparameters like $\delta$, $\tau$, and the choice of win probability thresholds? Third, does one global direction adequately represent the entire strategic landscape of chess, given how drastically positions differ between opening, middlegame, and endgame? The authors themselves note a failure case where all candidate moves are “winning,” and the projection metric provides no differentiation. These concerns suggest that the “single direction” assumption, while elegant, might be oversimplified.

*Issue 2*:
The introduction suggests that SOLIS achieves “2500+ Elo at 50 ms/move,” but this rating is derived entirely from matches against Stockfish configured with an internal Elo cap, rather than from a standard or independent rating pool. The reported Elo values are computed using BayesElo fits relative to that capped version of Stockfish, whose effective strength depends strongly on internal parameters (such as hash size, threads, contempt, and NNUE configuration). As a result, the “2500+ Elo” figure should be interpreted as relative performance under a constrained benchmark, not as an absolute or general rating. The framing in the introduction and abstract could therefore be made more precise to avoid overstating the engine’s general playing strength.

**Questions:**

* Clarification on the Elo framing: Could you clarify whether the “2500+ Elo” claim is explicitly relative to the capped Stockfish baseline (50 ms/move, 32 MB hash, etc.)? If not, please justify why this setup is a reasonable proxy for general playing strength.
* Phase dependence: Have you examined whether the projection scores correlate more strongly with Stockfish evaluations in specific phases of the game (e.g., openings vs. endgames)? A simple analysis using existing data could help evaluate whether the “one-vector” assumption holds consistently across phases.

*Minor comments*:
* The novelty is modest but elegant. It’s less about proposing a new architecture and more about demonstrating how latent geometry can effectively support planning in a complex, structured domain.
* The writing overall is clear, and figures are informative. Adding legends directly into the visualizations (e.g., color mappings in the latent space plots) would make them more self-contained.

---

### Official Review · Reviewer_Dgxn · 2025-11-10

**Soundness:** 2
**Presentation:** 2
**Contribution:** 2
**Rating:** 2
**Confidence:** 4

**Summary:**

This paper proposes SOLIS, a chess-playing system that removes conventional policy/value heads and instead performs planning via latent-space arithmetic. Specifically, it learns an evaluation-aligned embedding using supervised contrastive learning on Stockfish-labeled positions, defines a single global “advantage vector” from losing to winning examples, and performs shallow lookahead by selecting continuations whose latent vectors move most in this advantageous direction. The authors argue this constitutes a form of “latent planning” that is simpler and more interpretable than MCTS-based or policy/value-based systems. Experiments report >2500 Elo (vs. rating-limited Stockfish) at shallow depths (3–5), interpretability plots, and some ablations on search width and scoring variations.

The paper presents a creative and clearly written idea, but it does not yet reach the level of scientific rigor, novelty, and empirical validation expected for ICLR. The proposed approach can be viewed as a heuristic reformulation of value regression within a contrastive latent space, trained entirely on Stockfish-generated evaluations. While the idea of “latent planning” is intriguing, the work lacks theoretical depth and does not demonstrate clear novelty relative to established approaches such as value-based networks or metric-learning evaluation functions. Its dependence on a pre-existing engine for supervision and the limited, self-contained evaluation setup make it difficult to assess whether the observed performance reflects genuine planning ability or distilled Stockfish behavior. Overall, this is an interesting engineering exploration that shows how a contrastive embedding of engine evaluations can approximate its teacher with lower computational cost, but it falls short of offering a new algorithmic contribution or scientifically rigorous validation. Nonetheless, the perspective on representation-driven planning is thought-provoking and may inspire follow-up research toward more general and interpretable planning systems.

**Strengths:**

- Original conceptual framing: The paper introduces an appealing idea of planning as geometric movement within a learned evaluation-aligned space, offering a fresh perspective that reframes search as navigation in representation space rather than explicit value prediction.

- Interpretability and insight: UMAP visualizations and latent trajectories show that game progression corresponds to smooth motion along the advantage direction; decisive games exhibit coherent latent flow. This offers rare interpretability in game-playing models.

- Transparency and honesty: The authors clearly describe their pipeline (tokenization, SupCon training, advantage computation, search), disclose limitations (non-Markovian FEN, threefold repetition, mate-length), and promise code release. This makes the paper reproducible and practically valuable.

**Weaknesses:**

- Questionable novelty and conceptual framing: The “latent planning via advantage direction” boils down to using a linear classifier in embedding space that correlates with Stockfish’s scalar evaluation. Functionally, this is equivalent to training a value network and ranking continuations by predicted value — just with an extra cosine projection. The contrastive loss does not produce new algorithmic behavior, only a reparametrization of regression in latent space.

- The claim of planning in latent space is overstated: The search still performs an explicit breadth-depth expansion; no genuine planning mechanism (model rollout, learned dynamics, or policy reasoning) is introduced. “Global advantage direction” assumes a single monotonic dimension can represent chess advantage — a strong assumption that clearly fails in practice (positional sacrifices, tactical motifs). The paper provides no theoretical justification for why this should work beyond anecdotal success.

- Supervision and circularity: All supervision comes from Stockfish evaluations. The model effectively distils Stockfish into a smaller, less expressive student. Its apparent success against “Stockfish (capped)” is meaningless since both share the same evaluation function, but SOLIS searches fewer nodes and plays faster; its output is therefore trivially correlated with its teacher. This is a form of teacher-student distillation, not an independently motivated or novel planning system. Because the same engine defines both the training labels and the evaluation baseline (Elo against a variant of Stockfish), the results suffer from circular bias. It is impossible to tell whether SOLIS learned genuine positional understanding or just reproduces Stockfish’s biases.
This results in unreliable and non-standard evaluation. Elo ratings are reported only against a single opponent (Stockfish limited to 50 ms per move), not across a rating pool. No games are played against diverse engines (LCZero, Komodo, etc.), nor against human-established benchmarks. Elo under self-play or against a single fixed engine cannot be interpreted as absolute strength.
Moreover, there is no equal-time or equal-node comparison, which makes the “competitive” claim unsubstantiated. For instance, a depth-5 search with fixed width W = 8 may evaluate far fewer positions than the baseline Stockfish search — this difference is neither reported nor normalized.
Furthermore, the training/test split of chess positions is unclear, raising the possibility of data leakage (common openings seen during training re-appearing during play). This is critical, as a learned engine might simply memorize typical positions.

- Lack of ablations and diagnostics: The paper performs almost no serious ablations, no comparison to a simple value regression baseline trained with MSE to the same Stockfish scores, no analysis of the “advantage vector” sensitivity to percentile thresholds or batch sampling, no experiment removing the contrastive loss or replacing it with a simpler cosine regression — leaving unclear whether the proposed method offers any real benefit. Reported “anchored vs. unanchored” scoring differences are small (<20 Elo), within noise and no statistical tests or confidence intervals are reported.

- Weak empirical evidence of learning: The UMAP plots are mostly qualitative; they show smoothness but no quantitative measure of alignment between latent distance and game outcome. No training curves, loss diagnostics, or evidence of convergence are given. The only quantitative evidence of “understanding” is winning some games at shallow depth — but since the evaluation is derived from the teacher engine, this proves little about generalization.

- Poor scientific grounding: There is no theoretical analysis, no learning guarantees, no understanding of why the global vector should align with advantage. Terms like “planning”, “latent reasoning”, and “interpretability” are used loosely, often meaning “vector ranking”. Figures emphasize aesthetics over insight; no quantitative interpretability metric (e.g., rank correlation between latent cosine similarity and true outcome) is given.

Minor issues and clarity
- Hyperparameter justification or sensitivity curves (δ, τ, contrastive batch size) are absent.
- Some figures and tables lack standard deviations or error bars.

**Questions:**

- How do you ensure that the same positions or near-duplicates (via transpositions) from training do not appear in evaluation play?

- Why not train a scalar value head directly? What advantage does the cosine-based contrastive formulation provide?

- How sensitive are results to δ, τ, and the definition of “positive” pairs?

- What is the computational efficiency comparison at equal time or equal nodes against Stockfish NNUE?

- Have you tried using other engines’ evaluations as supervision? If so, how stable is the representation?

- Can you show that the latent distance structure correlates with true game outcomes (e.g., Spearman rank correlation vs. win probability)?

- How many distinct board positions were used for training, and are they balanced across phases (opening, middle, endgame)?

- Could the same principle be applied in continuous-control or Atari domains using learned evaluators (e.g., MuZero-style)? Atari games have partial observability, stochasticity, and non-Markov features (e.g., timers, hidden states). Many are not perfect-information turn-based games. The notion of “advantage direction” derived from static embeddings of frames might not translate directly!

- How sensitive is performance to the exact percentile used for the advantage vector (e.g., 99th vs 95th)?

- Would training on game outcomes rather than Stockfish evals yield similar structure?

- Can the authors comment on the computational efficiency (positions/sec) compared to baseline search?

---

### Meta-Review · Area_Chair_UV5F · 2025-12-19

**Summary:**

This paper proposes SOLIS, a chess-playing system that performs planning directly in a learned, evaluation-aligned embedding space. Using supervised contrastive learning on Stockfish-labeled positions, the method constructs a latent space in which outcome similarity corresponds to proximity and defines a single global “advantage vector” oriented from losing to winning positions. Planning is then carried out via shallow search by ranking candidate moves according to their alignment with this vector. Experiments in chess demonstrate competitive performance under constrained conditions (e.g., shallow search, short time controls), along with qualitative visualizations intended to support interpretability.

While the paper presents a clean, clearly written, and conceptually appealing idea, the reviewers consistently raise concerns regarding novelty, scientific rigor, and empirical validation. Taken together, these concerns outweigh the strengths of the submission. For these reasons, I recommend rejection.

**Reviewer Concerns:**

he reviewers agree that the paper presents a clean, well-written, and conceptually appealing idea: performing planning in chess by operating directly in an evaluation-aligned embedding space, using a single global advantage direction learned via supervised contrastive learning on Stockfish evaluations. The geometric framing and interpretability visualizations are generally viewed as the strongest aspects of the work, and several reviewers note that the simplicity of the method makes it easy to understand and potentially transferable.

However, there is broad consensus that the paper falls short in terms of novelty and scientific rigor. Reviewers consistently argue that the proposed method can be interpreted as a reparameterization of standard value-based approaches, effectively distilling Stockfish’s evaluation function into a latent space rather than introducing a genuinely new planning mechanism. The claim of “latent planning” is therefore seen as overstated, since the system still relies on explicit search and does not introduce learned dynamics or fundamentally new reasoning capabilities.

A major shared concern is the experimental evaluation. Because the model is trained entirely on Stockfish evaluations and evaluated primarily against a constrained version of Stockfish, the results suffer from circularity and are difficult to interpret. The reported Elo scores are relative to a specific baseline and short time control, and reviewers note the absence of comparisons to other engines, self-play calibration, equal-time or equal-node baselines, statistical significance analysis, and strong ablations. This makes it unclear whether the observed performance reflects genuine planning ability or simply efficient distillation of the teacher engine.

Reviewers also point to a lack of theoretical grounding and technical depth. The assumption that a single global advantage vector can capture progress across all phases of chess is seen as fragile and insufficiently justified, and known failure cases such as repetition and mate-length insensitivity underscore these limitations. Overall, while reviewers find the idea interesting and potentially inspiring for future work, they largely agree that the current evidence does not support the paper’s stronger claims, leading most to favor rejection or only a very weak acceptance.

**Reviewer Scores:**

The authors did not post a response

---

### Decision · Program_Chairs · 2026-01-26

Reject